# FCG-ASpredictor: An Approach for the Prediction of Average Speed of Road Segments with Floating Car GPS Data

**DOI:** 10.3390/s19224967

**Published:** 2019-11-14

**Authors:** Difeng Zhu, Guojiang Shen, Duanyang Liu, Jingjing Chen, Yijiang Zhang

**Affiliations:** 1College of Computer Science and Technology, Zhejiang University of Technology, Hangzhou 310023, China; zhudf@usx.edu.cn (D.Z.); ldy@zjut.edu.cn (D.L.); 2Yuanpei College, Shaoxing University, Shaoxing 312000, China; zhangyijiang@usx.edu.cn; 3Zhijiang College, Zhejiang University of Technology, Shaoxing 312000, China; jjchen@zjc.zjut.edu.cn

**Keywords:** intelligent transportation system, average speed prediction, GPS data, RLS-EKF algorithm

## Abstract

The average speed (AS) of a road segment is an important factor for predicting traffic congestion, because the accuracy of AS can directly affect the implementation of traffic management. The traffic environment, spatiotemporal information, and the dynamic interaction between these two factors impact the predictive accuracy of AS in the existing literature, and floating car data comprehensively reflect the operation of urban road vehicles. In this paper, we proposed a novel road segment AS predictive model, which is based on floating car data. First, the impact of historical AS, weather, and date attributes on AS prediction has been analyzed. Then, through spatiotemporal correlations calculation based on the data from Global Positioning System (GPS), the predictive method utilizes the recursive least squares method to fuse the historical AS with other factors (such as weather, date attributes, etc.) and adopts an extended Kalman filter algorithm to accurately predict the AS of the target segment. Finally, we applied our approach on the traffic congestion prediction on four road segments in Chengdu, China. The results showed that the proposed predictive model is highly feasible and accurate.

## 1. Introduction

The prediction of the average speed (AS) of road segments plays an important role in an intelligent transportation system (ITS). Its accuracy and timeliness have a great impact on the implementation of dynamic traffic management, such as traffic congestion estimation [1] and signal control [2]. The data collection of floating cars has the advantages of high flexibility, strong real-time performance, wide coverage, and high data precision, when compared to that of fixed detectors [3].

Existing researches usually relied on traffic parameters of fixed detectors to predict the AS in a road segment. The low accuracy is the main barrier for its wide application. Cetin and Comert [4] utilized the coil dataset published by California Path and then proposed the expectation maximization and Cumulative Sum (CUSUM) algorithms to predict the average traffic speed. Chandra and Al-Deek [5] mined the interaction between the upstream and downstream segments using dual-loop detector data and predicted the AS of road segments designed by a vector self-decreasing time series model. Jing et al. [6] assessed the multistep speed predictive performance of eight different models using 2-min road segment speed data collected from remote traffic microwave sensors. All above approaches failed to consider the traffic state of adjacent intersections, so it is difficult to accurately demonstrate the traffic state of urban roads via data acquisition from fixed detectors.

Recently, the booming of mobile Internet has inspired new ideas for traffic congestive detection. The mobile detective data from the vehicle have a wide coverage and continuous space, and the huge daily traffic data make the prediction of the traffic state more accurate and reliable. Emerging technologies based on Global Positioning System (GPS) enable us to track vehicle trajectories and collect real-time traffic data across entire road networks [7], and have been introduced to predict the AS of road segments. Queen and Albers [8] proposed a dynamic Bayesian model to identify lagged causal relationships between time series, and predict traffic speed at multiple road link locations. Pei et al. [9] collected GPS probed data of road segments, and developed a predictive model of AS using a full Bayesian method. Combining the acceleration of the target segment and the speed of the adjacent segment, Ye et al. [10] used an improved Neural Network (NN) to improve the prediction performance of AS. Based on a study of prediction bias correlation among adjacent road segments and weather factors, Yang et al. [11] employed an artificial NN and adjustment approach to predict the AS of a road segment. Yao et al. [12] developed a Support Vector Machine (SVM) model consisting of spatiotemporal parameters. It is commonly used for short-term prediction under the experimental condition that the runtime speed should be below 35 KM/h. Satrinia and Saptawati [13] combined map-matching with topological information to predict traffic speed via Support Vector Regression (SVR). Zhao et al. [14] adopted a deep learning model to predict the traffic speed during non-recurrent congestion periods. These approaches perform well only if the GPS data sampling is sufficient. On the other hand, the predictive accuracy of these approaches based on NNs, SVM, and SVR usually depend on the training quality of the traffic dataset.

Apart from the aforementioned road traffic predictions, Kalman filter (KF) does not depend on the training quality of the traffic dataset, and is one of the most widely used traffic prediction methods, which was first introduced in traffic forecasting by Okutani and Stephanedes [15]. KF addressed the problem of filtering the recursion of discrete linear data, which is applied to the fields of traffic variable prediction and travel time estimation [16,17]. However, due to its linear model, it is not appropriate for nonlinear and random traffic variables. To overcome this issue, an extended Kalman filter (EKF) that is suitable for nonlinear traffic prediction is implemented with the KF algorithm, which linearizes the nonlinear state space model. Liu et al. [18] proposed a state-space model and a progressive EKF method. It fuses heterogeneous data and tracks the variation in traffic dynamics. Yuan et al. [19,20] later used the EKF to predict the traffic states, in which the discretized Lagrangian model was used as the process equation. Based on the EKF, Dong et al. [21] developed a spatiotemporal model to predict traffic flow. Huang et al. [22] designed an advanced EKF algorithm to improve the accuracy of vehicle speed prediction by combining the adaptive forgetting factor and the EKF algorithm. Although EKF has been widely adopted to speed prediction, it failed to enable high accuracy and parameter estimation, as well as random factors.

Recursive least squares (RLS) is used to correct the previous results by using new observational data. RLS usually performs real-time traffic state estimation toward the system parameters [23]. Comert et al. [24] adopted a RLS filtering and proposed a model for predicting traffic speed with the considerations to impact factors such as weather, accidents, and driving characteristics. A weighted RLS estimator was used to optimize these parameters of the linear functions. Tang et al. [25] established the Takagi–Sugeno-type fuzzy rules to forecast travel speed. Aiming to optimize wireless network performance, Kulkarni et al. [26] proposed a simple traffic mechanism to predict traffic load by using RLS. However, RLS performs poor recognition accuracy if noise exists.

Hybrid models incorporate the advantages of single approaches to improve traffic prediction accuracy [27,28]. However, the road segment data used [27] is not sufficient, and random events should be taken into account for further accuracy improvement.

Recently, existing researches based on motion detectors have had these problems. On one hand, the accuracy of AS prediction would be affected when the road segment data is not sufficient or a random event occurs. On the other hand, the predictive accuracy of Machine Learning methods such as NNs, SVM, and SVR usually depend on the training quality of the dataset.

In this study, we proposed a novel road segment AS prediction model based on floating car GPS data (FCG-ASpredictor), which adopted a spatiotemporal correlation calculation method and a recursive least squares–extended Kalman filter (RLS-EKF) to solve current issues. Finally, we identified our approach on the AS prediction on four road segments in Chengdu and found that FCG-ASpredictor is feasible and highly accurate.

The rest of the paper is organized into five sections. Section 2 analyzes the data association. Section 3 describes the materials and methods. Section 4 illustrates the experimental results. Section 5 discusses the evaluation and feasibility. Finally, conclusions are drawn in Section 6.

## 2. Data Association Analyses

Based on the GPS data of floating cars in Chengdu in November 2016, we adopted a K-means Clustering algorithm to calculate the frequency distribution. Based on the frequency-intensive areas of GPS data and historical data, we also used the Pearson correlation coefficient [29] to analyze the correlation of the AS. In addition, we analyzed the impact of other sudden factors (such as weather, date attributes, etc.) on the AS of road segments.

### 2.1. Historical Data Correlation Analyses

A traffic dataset that contains time-series data is chronologically consecutive. As a typical time-series set, the AS of a road segment is analyzed on an hourly basis, with which we can comprehensively study its internal relationship. The current time interval is closely related to the AS of the adjacent road segments. As shown in Figure 1, the AS of the north third section of the First Ring Road for the time span (1 November to 7 November 2016) is selected. In addition to an obvious sudden change in the AS during the traffic rush hours, the data correlation between these two adjacent timeslots is large, and the trend of change is coherent.

A correlation coefficient analysis is a statistical method that reflects the close relationship between variables [30] and can be used to reveal the degree of influence on traffic conditions during adjacent hours. The Pearson correlation coefficient is a measure of the strength of a linear relationship between two variables. In this study, the Pearson correlation coefficient was used to analyze the correlation between the AS of the road segment during these adjacent timeslots.

The AS of 500 road segments in the main area is divided to 24 timeslots during November. The average Pearson correlation coefficient is formulated as follows:(1)$$Pe(t,t+1)=\frac{Cov(X_{t},X_{t+1})}{\sqrt{Var(X_{t})Var(X_{t+1})}}$$
where Pe(t,t+1) is the Pearson correlation coefficient for adjacent timeslots, X is the AS dataset of all of the same timeslots over 30 days, $Cov(X_{t},X_{t+1})$ is the AS covariance of road segments in adjacent timeslots within 30 days, and Var(·) is the variance.

As shown in Figure 2, the Pearson correlation coefficient between adjacent hours in the range of 24 timeslots is positive—that is, the current AS of the road segment has a correlation with the forward timeslot under normal conditions. According to the aforementioned analysis, the AS of the current timeslot is correlated with the forward timeslot, but the correlation is not large when traffic rush hours are encountered. Therefore, the AS prediction only considers the data value of the forward timeslot, leading to a low accuracy. The historical data of different timeslots are an important component of the AS prediction of the road segment. By comprehensively analyzing the influencing weights of different timeslots in the historical AS data, the accuracy of the AS prediction can be improved.

The historical influencing factor includes forward timeslot data and historical simultaneous timeslot data—that is, the AS of the previous six timeslots and the AS of the previous seven days of historical timeslots, respectively. The Pearson correlation coefficient of the 13 influencing factors in each road segment is calculated. Take the four road segments in Chengdu, China as an example. Their Pearson correlation coefficients are listed in Table 1, indicating that the AS of the road segment is closely related to the AS of the forward timeslots and the AS of the historical simultaneous timeslots. The AS of the four road segments for the previous 1–6 h and the first, sixth, and seventh historical simultaneous timeslots are positively correlated. Meanwhile, the AS correlation degree of the forward timeslot decreases with time, which indicates that the above nine influencing factors are considerations of the AS prediction of the four target segments. The degree of correlation varies with the road segment and timeslot.

### 2.2. Correlation Analyses of Other Factors

It can be seen from the foregoing correlation analysis that the traffic dataset changes in chronological order and has coherence, and the historical simultaneous timeslot data and the forward timeslot data have different degrees of influence on the current timeslot data. However, the daily traffic status does not completely obey the normal law of historical data. When affected by external dynamic factors such as weather, date attributes, and emergencies, the traffic status may cause special circumstances, making the traffic situation deviate from the long-term trend [31,32]. Emergencies have greater randomness and unpredictability, and the corresponding datasets are limited. Therefore, this work mainly analyzes the influence of other external factors such as weather and date attributes on the traffic road segment speed.

Rain and snow worsen the road conditions and gradually result in traffic congestion, as shown in Figure 3. Since the period of the sixth day to the ninth day was rainy in November, the 16 days of each hour of the road segment AS in November were selected (the four days of the sixth day to the ninth day had light rain, and the remaining 12 days were cloudy or sunny under the same week attribute). The abscissa of Figure 3 represents the AS of the road segment over 24 h (km/h), and the ordinate indicates the date in different types of weather. The deeper the red color, the slower the AS. Meanwhile, the darker the blue color, the faster the AS. It is clear that the AS during the light rainy days is basically slower than that of other days, so the influence of external factors such as the weather on traffic congestion cannot be ignored.

Figure 4 demonstrates that the AS of a road segment is different in the state presented on weekdays and weekends. The abscissa represents the AS of a road segment over 24 h each day (km/h), and the ordinate represents the date of two consecutive weeks (seventh day to 11th day and 14th day to 18th day are the weekdays; and the 12th day, 13th day, 19th day, and 20th day are the weekends) in the figure. This indicates that the phenomenon of morning and evening rush hours is brighter on weekdays, while it is weakened during the weekend. This is clearly related to people’s travel behavior: people need to go to work on weekdays, and they travel less on weekends.

## 3. Materials and Methods

The traffic flow system is a highly correlated system, and a change is random at a certain moment, which makes traffic status prediction difficult. RLS can realize the real-time estimation of system parameters and has a great influence on model identification accuracy under noisy conditions. An EKF can be applied to nonlinear system prediction, but it is susceptible to the accuracy of the state estimation.

In order to compensate for the defects of the respective methods and solve the issue of insufficient road segment data, the main idea of FCG-ASpredictor is shown in Figure 5. By establishing multiple regression equations, the historical AS obtained by the spatiotemporal correlation calculation method and the external factors (i.e., weather and date attribute) of the current timeslot are identified by the RLS. The measured values and observed values are adopted by the EKF to improve the predictive accuracy of the AS of the target road segment.

### 3.1. Study Area and Data Sources

Chengdu, as the capital of Sichuan Province in China, is an important central city in the western region. Its geographical coordinate range is 30^◦^05′–31^◦^26′ latitude and 102^◦^54′–104^◦^53′ longitude. It is consist of 20 districts, covering the total area of 14,335 km^2^, with a resident population of 16.33 million. This paper selected the central urban areas, the Wuhou, Jingjiang, Qingyang, Jinniu, and Chenghua districts, as the study areas.

Due to the high sampling frequency of floating cars data, we employ the dataset (i.e., order details) from the Chengdu branch of Didi Chuxing, The sampling frequency is 3 s. The data size is 462 GB, and each record includes: (1) driver ID; (2) order ID; (3) timestamp; (4) latitude; (5) longitude; and (6) vehicle status. The raw data format is shown in Table 2.

### 3.2. The Computational Procedures of AS

The AS of the road segment usually refers to the AS of travel through the road segment. We employ the travel speed of the road segment by using the accumulated integral of the instantaneous speed, and obtain the AS of the road segment.

According to the position and timestamp of the adjacent position belonging to the same order ID, the distance between adjacent positions can be calculated by using the spherical distance formula. The time interval can be calculated by the timestamp of the adjacent positions. The instantaneous speed of each position is calculated as follows:(2)$$v=\frac{r*\arccos\{sin(x_{1})*\sin(x_{2})+\cos(x_{1})*\cos(x_{2})*\cos(y_{1}-y_{2})\}}{\left|T_{1}-T_{2}\right|}$$
where *v* is the instantaneous speed, *r* is the earth radius, *x_1_* and *x_2_* are the latitudes of the adjacent positions, *y_1_* and *y_2_* are the longitudes of the adjacent positions, and *T_1_* and *T_2_* are the time stamps of the adjacent positions.

The travel distance of a positioning car based on the accumulated integral is calculated as follows:
(3)$$d_{tra}=\int_{t_{0}}^{t_{k}}vdt\approx v_{0}(\frac{t_{1}-t_{0}}{2})+\sum_{i=1}^{k-1}v_{i}\left(\frac{t_{i+1}-t_{i-1}}{2}\right)+v_{k}(\frac{t_{k}-t_{k-1}}{2})$$
where $d_{tra}$ is the travel distance, t(·) is the GPS positioning time, and v(·) is the instantaneous speed.

Since the sampling frequency is fixed, Formula (3) is modified as follows:(4)$$d_{tra}=t_{\mathrm{int}}(\frac{v_{0}}{2}+\sum_{i=1}^{k-1}v_{i}+\frac{v_{k}}{2})$$
where $t_{\mathrm{int}}$ is the fixed time interval.

According to the travel distance and time interval, the travel speed is calculated as follows:(5)$$v_{tra}=\frac{d_{tra}}{t_{k}-t_{0}}$$
where $v_{tra}$ is the travel speed.

Owing to the uneven distribution of the floating car in the urban road network, the speed measurement accuracy is degraded, and the AS prediction of the road segment is considered from the distribution of the floating car. In order to ensure accurate calculation of the AS of the road segment, the number of travel speed samples *n* at a certain time should not be less than the minimum number of samples *n_min_*. If the number of travel speed samples *n* is insufficient, then the historical AS and AS of the upstream and downstream segments during the simultaneous timeslot need to be integrated.

In addition, if the cumulative number *m* of continuous travel speed samples is greater than the maximum value *m_max_*, this indicates that the number of travel speed samples in the previous *m_max_* timeslots is continuously less than the minimum number of samples *n_min_*, and the AS of the upstream and downstream segments in the simultaneous timeslot is insufficient to reflect the current traffic status. Then, it is necessary to integrate the historical AS of road segments. The spatiotemporal correlation calculation process of AS is shown in Figure 6.

The formula for calculating the AS of the road segment is as follows:(6)$$\bar{v}(t)=\sum_{i=1}^{n}v_{tra_{i}}(t)/n$$
where $\bar{v}(t)$ is the AS of the road segment during timeslot *t*, *n* is the number of travel speed samples, and $v_{tra_{i}}(t)$ is the *i*th travel speed at timeslot *t*.

If the travel speed sample number *n* of the road segment at timeslot *t* is smaller than the minimum sample number *n_min_*, then the historical AS and the simultaneous AS of the upstream and downstream segments are integrated as follows:(7)$$\begin{array}{l} \bar{v}(t)=(1-\theta )\frac{1}{n}\sum_{i=1}^{n}v_{tra_{i}}(t)+\theta \left[\left(1-\omega \right){\bar{v}}_{h}(t)+\omega {\bar{v}}_{a}(t)\right]\\ \theta =\{\begin{array}{cc} 1-\frac{n}{n_{\min}}, & n\leq n_{\min}\\ 0, & n>n_{\min} \end{array}\\ \omega =\{\begin{array}{cc} 1-\frac{m}{m_{\max}}, & m<m_{max}\\ 0, & m\geq m_{\max} \end{array} \end{array}$$
where ${\bar{v}}_{h}(t)$ is the estimated historical AS of the road segment, and ${\bar{v}}_{a}(t)$ is the estimated AS of the upstream and downstream segment during the current timeslot. The control parameters *n_min_* and *m_max_* are derived from the example calibration.

${\bar{v}}_{h}(t)$ and ${\bar{v}}_{a}(t)$ are calculated by weighting the corresponding correlation speeds. The weighting formula is as follows:(8)$${\bar{v}}_{h}(t)=\alpha {\bar{v}}_{h}^{\prime }(t)+(1-\alpha ){\bar{v}}_{h}^{\prime }(t-1)$$
(9)$${\bar{v}}_{a}(t)=\beta {\bar{v}}_{u}(t)+(1-\beta ){\bar{v}}_{d}(t)$$
where ${\bar{v}}_{h}^{\prime }(t)$ and ${\bar{v}}_{h}^{\prime }(t-1)$ are the AS of the historical simultaneous timeslot and the AS of the forward timeslot, respectively; ${\bar{v}}_{u}(t)$ and ${\bar{v}}_{d}(t)$ are the AS of the upstream and downstream segments during the current timeslot, respectively; and α and β are weight coefficients that are adjusted according to the measurement of actual data.

### 3.3. Establishment of Multiple Regression Equations

According to the impact of historical AS, weather, and date attributes on AS prediction, the degree of influence between the AS of the target road segment and the historical AS is calculated by the Pearson correlation coefficient.

The AS in the historical simultaneous timeslots of the previous $n_{t}$ days, the AS of the previous $n_{p}$ timeslots during a day, the weather value of the current timeslot, and the date attribute value of the current timeslot are selected. The following multiple regression equation for predicting the AS value is established:(10)$${\bar{v}}_{k}\left(t\right)=\left[\begin{array}{cccc} a_{1} & a_{2} & \cdots & a_{n_{t}} \end{array}\right]\left[\begin{array}{c} {\bar{v}}_{k-1}\left(t\right)\\ {\bar{v}}_{k-2}\left(t\right)\\ \vdots \\ {\bar{v}}_{k-n_{t}}\left(t\right) \end{array}\right]+\left[\begin{array}{cccc} b_{1} & b_{2} & \cdots & b_{n_{p}} \end{array}\right]\left[\begin{array}{c} {\bar{v}}_{k}(t-1)\\ {\bar{v}}_{k}(t-2)\\ \vdots \\ {\bar{v}}_{k}(t-n_{p}) \end{array}\right]+\left[\begin{array}{cc} c_{1} & c_{2} \end{array}\right]\left[\begin{array}{c} x_{k,1}\left(t\right)\\ x_{k,2}\left(t\right) \end{array}\right]$$
where ${\bar{v}}_{k}\left(t\right)$ is the predicted AS during timeslot *t* of the *k*th day, ${\bar{v}}_{k-1}\left(t\right),\ldots ,{\bar{v}}_{k-n_{t}}\left(t\right)$ are the AS in the historical simultaneous timeslot *t* of the previous $n_{t}$ days, and ${\bar{v}}_{k}(t-1),\ldots ,{\bar{v}}_{k}(t-n_{p})$ are the AS in the previous $n_{p}$ timeslots of the *k*th day. $x_{k,1}\left(t\right)$ and $x_{k,2}\left(t\right)$ are the weather-quantized value and the date-attribute-quantized value, respectively, in timeslot *t* of the *k*th day; these need to be quantified according to the standard. $a_{1}\ldots a_{n_{t}}$,$b_{1}\ldots b_{n_{p}}$,$c_{1}$ and $c_{2}$ are the influence weights of each system variable on the predicted value.

### 3.4. System Identification of RLS Method

The system parameters are identified and updated according to Formula (10). The transformed recursive equation is as follows:(11)$${\bar{v}}_{k}\left(t\right)=\varphi_{k}{}^{T}(t)\theta +e_{k}\left(t\right)$$
where $\varphi_{k}{}^{T}$ is the AS value of the road segment, θ is the identified parameter vector, and $e_{k}\left(t\right)$ is the error caused by observation noise. $\varphi_{k}{}^{T}$ and θ are recorded as vectors as follows:(12)$$\varphi_{k}{}^{T}(t)=\left[{\bar{v}}_{k-1}\left(t\right),\ldots ,{\bar{v}}_{k-n_{t}}\left(t\right),{\bar{v}}_{k}(t-1),\ldots ,{\bar{v}}_{k}(t-n_{p}),x_{k,1}\left(t\right),x_{k,2}\left(t\right)\right]$$

(13)$$\theta ={\left[a_{1},\ldots ,a_{n_{t}},b_{1},\ldots b_{n_{p}},c_{1},c_{2}\right]}^{T}.$$

Combining Formulas (11), (12), and (13), the system parameter identification gain and the error covariance matrix are updated. The least-squares equation is expressed as follows:(14)$$K_{k}(t)=P_{k}(t-1)\varphi_{k}(t){\left[\varphi_{k}{}^{T}(t)P_{k}(t-1){\hat{\theta }}_{k}(t-1)\varphi_{k}(t)+1\right]}^{-1}$$
(15)$$P_{k}\left(t\right)=\left[I-K_{k}\left(t\right)\varphi_{k}{\left(t\right)}^{T}\right]P_{k}\left(t-1\right)$$
where $K_{k}(t)$ is the parameter identification gain for timeslot *t*, $P_{k}(\cdot )$ is the error covariance matrix of different timeslots, and I is the identity matrix of the identification parameter.

According to Formulas (6), (7), and (11) to (15), the recursive formula for system parameter identification during timeslot *t* is expressed as follows:(16)$${\hat{\theta }}_{k}\left(t\right)={\hat{\theta }}_{k}\left(t-1\right)+K_{k}\left(t\right)\left[{\bar{v}}_{}\left(t-1\right)-\varphi_{k}^{T}\left(t-1\right){\hat{\theta }}_{k}\left(t-1\right)\right]$$
where ${\hat{\theta }}_{k}(\cdot )$ is the least-squares estimate of the system parameters for different timeslots, and ${\bar{v}}_{}\left(t-1\right)-\varphi_{k}^{T}\left(t-1\right){\hat{\theta }}_{k}\left(t-1\right)$ is the correction term of the identified parameter estimation for timeslot *t*−1.

### 3.5. Implementation of EKF

It can be seen from Formula (10) that the AS prediction model includes nonlinear external factors such as the weather and date attributes. This study uses an EKF algorithm to improve the AS prediction accuracy of the target segment. For the sake of simplicity, Formula (10) is modified as follows:(17)$$x\left(t\right)=\left[\begin{array}{cccc} a_{1} & a_{2} & \cdots & a_{n_{t}} \end{array}\right]\left[\begin{array}{c} x\left(t-\Delta \right)\\ x\left(t-2\Delta \right)\\ \vdots \\ x\left(t-n_{t}\Delta \right) \end{array}\right]+\left[\begin{array}{cccc} b_{1} & b_{2} & \cdots & b_{n_{p}} \end{array}\right]\left[\begin{array}{c} x(t-1)\\ x(t-2)\\ \vdots \\ x(t-n_{p}) \end{array}\right]+\left[\begin{array}{cc} c_{1} & c_{2} \end{array}\right]\left[\begin{array}{c} x_{k,1}\left(t\right)\\ x_{k,2}\left(t\right) \end{array}\right]$$
where Δ is the number of timeslots in a day (assuming the length of the timeslot and the number of timeslots remain constant), x(t) is the AS prediction of the road segment, $x\left(t-\Delta \right),\cdots ,x\left(t-n_{t}\Delta \right)$ are the AS in the historical simultaneous timeslots of the previous $n_{t}$ days, and $x(t-1),\cdots ,x(t-n_{p})$ are the AS of the previous $n_{p}$ timeslots during a day.

According to Formula (17), the standard form of the state equation and the observation equation are expressed as follows:(18)$$\begin{array}{c} x(t)=f\left(X(t-1)\right)+w(t-1)\\ y(t)=g\left(X(t)\right)+m(t) \end{array}$$
where x(t) and y(t) are state and observation vector values, respectively; w(t−1) is the system process noise; m(t) is the observation noise; and f(X(t−1)) and g(X(t)) are nonlinear mapping functions of the state equations and observation equations, respectively. X(t−1), f(X(t−1)), and g(X(t)) are expressed as follows:(19)$$X(t-1)={\left[\begin{array}{cccccccc} x\left(t-\Delta \right) & \cdots & x\left(t-n_{t}\Delta \right) & x(t-1) & \cdots & x(t-n_{p}) & x_{k,1}\left(t\right) & x_{k,2}\left(t\right) \end{array}\right]}^{T}$$
(20)$$f\left(X(t-1)\right)\approx f\left(\overset{⌢}{X}(t-1)\right)+A(t-1)\left(X(t-1)-\overset{⌢}{X}(t-1)\right)$$
(21)$$g\left(X(t)\right)\approx g\left(\overset{⌢}{X}(t)\right)+B(t)\left(X(t)-\overset{⌢}{X}(t)\right)$$
where $\overset{⌢}{X}$ is the estimated value of X, and A and B are the system state matrix and the observation matrix, respectively.

According to Formulas (17) to (21), A and B are derived as follows:(22)$$A(t-1)=\frac{\partial f(\overset{⌢}{X}(t-1))}{\partial X(t-1)}$$
(23)$$B(t)=\frac{\partial g(\overset{⌢}{X}(t))}{\partial X(t)}.$$

The three components of state vector X(t−1) in Formula (19) are x(t−n), $x_{k,1}(t)$, and $x_{k,2}(t)$. They are partial derivatives. A and B are converted to a Jacobian matrix:(24)$$\begin{array}{c} A(t-1)=[A_{1}\;A_{2}\;A_{3}]\\ B(t)=[1\;0\;0] \end{array}.$$

The corresponding parameters $A_{1}$, $A_{2}$, and $A_{2}$ in Formula (24) are calculated as follows:(25)$$\begin{array}{c} A_{1}={\left[a_{1},\ldots a_{n_{t}},b_{1},\ldots b_{n_{p}}\right]}^{T}\\ A_{2}=c_{1}\\ A_{3}=c_{2} \end{array}.$$

Since the specific values of parameters $A_{1}$, $A_{2}$, and $A_{3}$ corresponding to Formula (25) are calculated by Formula (16), then A(t−1) and B(t) are known values. Combining with the KF, the time update of Formula (17) is expressed as follows:(26)$$X{\left(t\right)}^{-}=f\left(X(t-1)\right)$$
(27)$$P{\left(t\right)}^{-}=A(t-1)P{\left(t-1\right)}^{+}A{\left(t-1\right)}^{T}+Q$$
where $X{\left(t\right)}^{-}$ is the prior estimate of the state vector at timeslot *t*, P is the covariance of the state vector estimation error, and Q is the covariance matrix of the process noise.

According to Formulas (18), (23), (26), and (27), the observation update of Formula (17) is expressed as follows:(28)$$G\left(t\right)=P{\left(t\right)}^{-}B{\left(t\right)}^{T}{\left(B\left(t\right)P{\left(t\right)}^{-}B{\left(t\right)}^{T}+R\right)}^{-1}$$
(29)$$P{\left(t\right)}^{+}=(I-G(t)B(t))P{\left(t\right)}^{-}$$
(30)$$X{\left(t\right)}^{+}=X{\left(t\right)}^{-}+G(t)(Y(t)-g(X{\left(t\right)}^{-}))$$
where G(t) is the Kalman gain at timeslot *t*, $X{\left(t\right)}^{+}$ is the posterior estimate of the state vector at timeslot *t*, and R is the covariance matrix of the observation noise.

## 4. Results

Since traffic control and guidance require real-time prediction, the length of the traffic prediction horizon is short, usually no more than 1 h. In this study, the prediction horizons are set to 15 min, 30 min, and 1 h, respectively. All experiments are compiled and tested based on Python 3.7 and TensorFlow 1.13.1.

### 4.1. Data Preprocessing

In order to make the selected segments more objectively reflect the advantages of our approach, four road segment speed datasets were adopted under the different road types. The road segment information is shown in Table 3. According to the characteristics of the Chengdu urban network, the segments 01_521, 03_6479, 04_6276, and 06_28250 belong to the main urban road, general road, ring road, and outer ring road, respectively. The time span of these datasets is from June to November 2016. In order to be consistent with the comparison algorithm long short term memory–recurrent neural network (LSTM-RNN) [27] and autoregressive integrated moving average model- Kalman filter (ARIMA-KF) [28], we select the AS data from 1 June 2016 to 31 October 2016 as the training sample and the AS data from 1 November to 30 November 2016 as the forecast sample.

To solve the issue of data drift, we use map-matching technology to obtain a standard dataset. In terms of the driving characteristics of the floating car, the DiDi cars do not represent the normal traffic state of the road segment under the vehicle status of empty and parking. Thus, we remove the records of the vehicle status of empty and parking.

The datasets include the AS of the road segments [derived from Formulas (2) to (9)], and the quantified values of external factors such as the weather and date attributes. According to the degree of external factors affecting traffic flow [33], the weather and date attributes are quantified as shown in Table 4.

### 4.2. Result Analysis

Corresponding to the data in Table 4, multivariate regression equations are established by taking the correlation factor values and the AS of four target road segments, respectively. Identified by the RLS method and considering nonlinear external factors such as weather and date attributes, the EKF algorithm is used to predict the AS for the current timeslot. Root mean square error (RMSE), mean absolute error (MAE), and mean absolute percentage error (MAPE) are used as the evaluation metrics. Figure 7, Figure 8, Figure 9, Figure 10, Figure 11, Figure 12, Figure 13, Figure 14 and Figure 15 show that the AS predictions based on the RLS-EKF are superior to those according to other two algorithms. All evaluation values of AS predicted on three different prediction horizons based on the RLS-EKF are lower than those based on the other two algorithms, which means the accuracy and stability of AS predicted on three different prediction horizons based on the RLS-EKF are superior to those based on the other two algorithms for the four road segments.

## 5. Discussion

### 5.1. Evaluation

The comparative analysis of proposed algorithms and existing algorithms are performed by three commonly used metrics in traffic prediction, including (1) RMSE, (2) MAE, and (3) MAPE. The three evaluation metrics are defined as follows:(31)$$RMSE=\sqrt{\frac{1}{n}\sum_{i}^{n}{\left(\hat{v}\left(t_{i}\right)-\bar{v}\left(t_{i}\right)\right)}^{2}}$$
(32)$$MAE=\frac{1}{n}\sum_{i}^{n}\left|\hat{v}\left(t_{i}\right)-\bar{v}\left(t_{i}\right)\right|$$
(33)$$MAPE=\frac{1}{n}\sum_{i}^{n}\frac{\left|\hat{v}\left(t_{i}\right)-\bar{v}\left(t_{i}\right)\right|}{\bar{v}\left(t_{i}\right)}$$
where $\hat{v}\left(t_{i}\right)$, $\bar{v}\left(t_{i}\right)$ are the predicted value and estimated value at timeslot $t_{i}$, respectively, and *n* is the timeslot number.

The estimated value is a relative value that is obtained from the historical AS. To predict the AS of the current timeslot t, the AS in the historical simultaneous timeslots of the previous n_t_ days, the AS of the previous n_p_ timeslots, the weather value of the current timeslot, and the date attribute value of the current timeslot are selected. When entering the next timeslot, the AS of the timeslot t is calculated as the estimated value according to Formulas (6) to (9).

From Table 5, RLS-EKF achieves the better performance with all three metrics for all prediction horizons, and the advantage becomes more evident in the four road segments.

All metrics of RLS-EKF are lower than those based on the other two algorithms (LSTM-RNN and ARIMA-KF). LSTM-RNN and ARIMA-KF are the latest traffic prediction approaches, and the difference between all three evaluation metrics of RLS-EKF and those based on other two algorithms are larger, which means that the experiments based on RLS-EKF have achieved good results.

From the perspective of three prediction horizons, all three metrics increase as the prediction horizon increases. The difference between all three evaluation metrics of one-h intervals and those of 30-min intervals are larger than the difference between those of 30-min intervals and 15-min intervals, which means that long-term traffic forecasting needs to consider more alternative influencing factors for optimization.

From the perspective of four road segments, the errors of road segment 01_521 and 04_6276 are larger than those of other two road segments.

### 5.2. Feasibility

The multiple regression equations in Section 3.3 contain four factors: (1) the AS of the historical simultaneous timeslot (AS-hst); (2) the AS of the forward timeslot (AS-ft); (3) the weather condition of the current timeslot (WC-ct); and (4) the date attribute of the current timeslot (DA-ct). In order to demonstrate the influence of four factors on the AS prediction, according to the equations in Section 3.3, we select five influencing cases, which respectively leave out the AS of the historical simultaneous timeslot (Miss-AS-hst), the AS of the forward timeslot (Miss-AS-ft), the weather condition of the current timeslot (Miss-WC-ct), the date attribute of the current timeslot (Miss-DA-ct), and lastly, do not have any missing factor.

The RMSE of five different influencing cases based on RLS-EKF is illustrated in Table 6. We further analyzed the feature contributions of five different influencing cases toward four road segments for three predicted dimensions. All the RMSEs of the no missing factor case are lower than those of four missing factor cases such as Miss-AS-hst, Miss-AS-ft, Miss-WC-ct, and Miss-DA-ct in the same road segment and predicted horizon, which means that the four factors of the equations in Section 3.3 are contributed to improve the predicted accuracy. In addition, Miss-AS-hst is the highest, Miss-AS-ft is the second highest, Miss-WC-ct is the third highest, and Miss-DA-ct is the lowest in the RMSE comparison of four missing factor cases. That means that AS-hst is the largest, AS-ft is the second largest, WC-ct is the third largest, and DA-ct is the smallest regarding the feature contributions of the predicted accuracy.

To improve the predicted accuracy of the FCG-ASpredictor, it is reasonable and feasible to select AS-hst, AS-ft, WC-ct and DA-ct as import impact factors of the equations in Section 3.3.

## 6. Conclusions

In this paper, we propose an integrated analysis model of predicting road segment AS: FCG-ASpredictor. It incorporates the spatiotemporal correlation calculation and RLS-EKF to address two issues: (1) low accuracy due to insufficient data and (2) poor training quality. By using traffic data in Chengdu, China to verify the proposed model, the analysis result is feasible. The main contributions of this paper are as follows: (1) new design to obtain an accurate AS of the road segment: we use the number of travel speed samples and the cumulative number of segments with less continuous travel speed samples as the benchmark metrics, and build a spatiotemporal correlations calculation method with regard to GPS data; (2) new approach based on RLS-EKF, which utilizes the RLS to fuse the historical AS with other factors (such as weather and date attributes) and apply EKF to predict the AS in the target segment. The experimental result shows that the RLS-EKF performs well and achieves high accuracy.

The FCG-ASpredictor combines various impact factors such as AS-hst, AS-ft, WC-ct, DA-ct, etc., and achieves good results for the AS prediction of road segments. However, there still exists limitations while applying the model for the speed prediction of long-term traffic; thus, we will work toward improving the model adaptation on spatiotemporal correlations in the future.

## Figures and Tables

**Figure 1 sensors-19-04967-f001:**
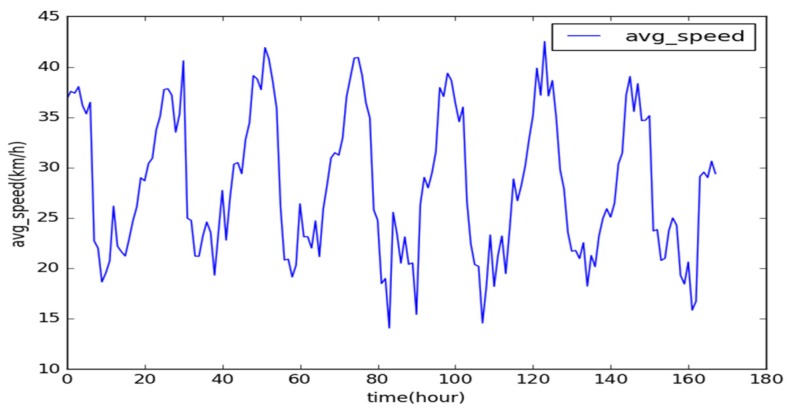
Average speed (AS) time-series data for the first seven days of November (in h).

**Figure 2 sensors-19-04967-f002:**
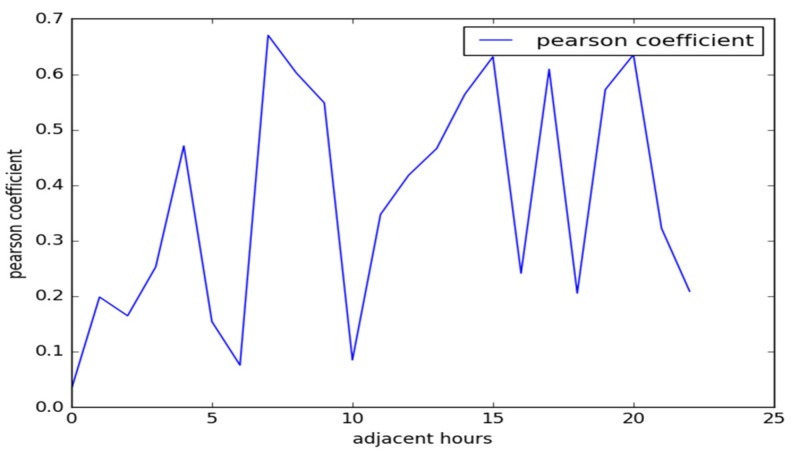
Average Pearson correlation coefficient of AS between adjacent hours.

**Figure 3 sensors-19-04967-f003:**
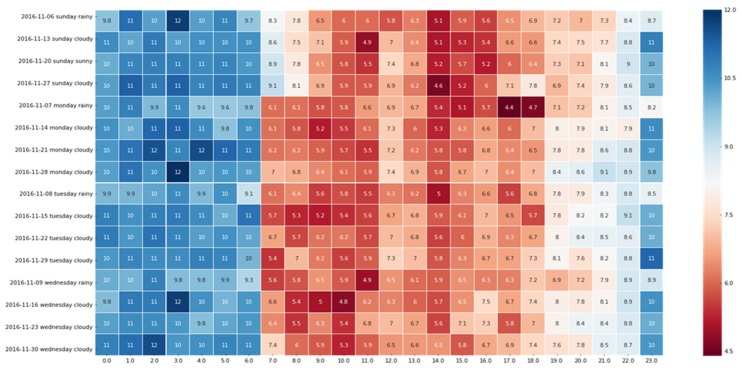
AS condition under different weather conditions.

**Figure 4 sensors-19-04967-f004:**
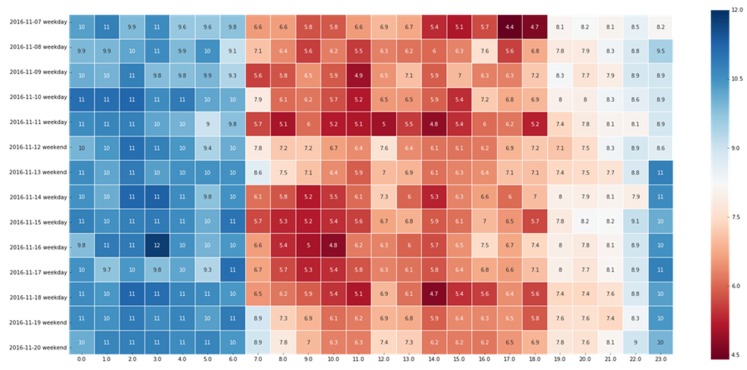
AS condition over two consecutive weeks.

**Figure 5 sensors-19-04967-f005:**
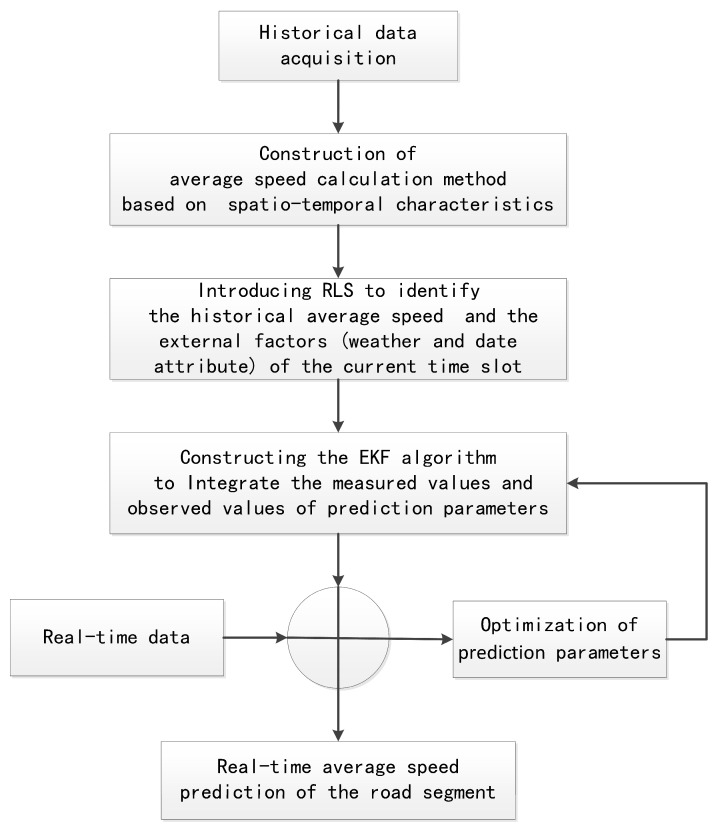
Our novel road segment AS prediction model based on floating car Global Positioning System data (FCG-ASpredictor) based on recursive least squares–extended Kalman filter (RLS-EKF).

**Figure 6 sensors-19-04967-f006:**
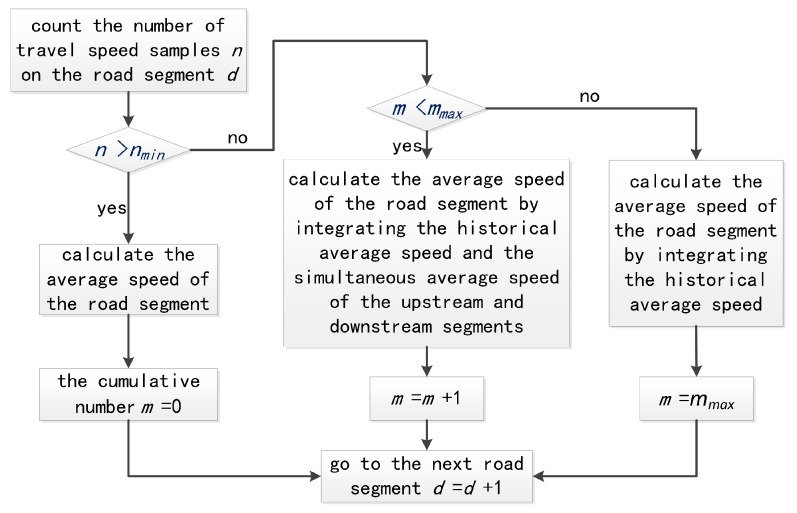
The process of AS calculation.

**Figure 7 sensors-19-04967-f007:**
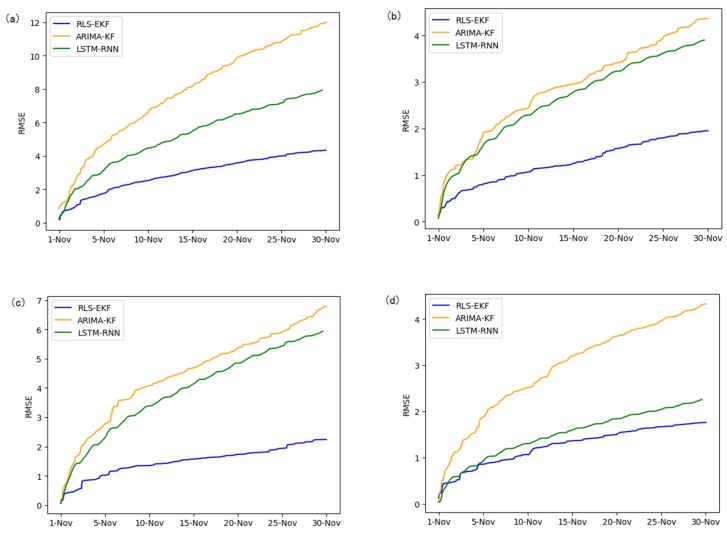
Root mean square error (RMSE) of 01_521 (**a**), 03_6479 (**b**), 04_6276 (**c**), and 06_28250 (**d**) at 15-min intervals.

**Figure 8 sensors-19-04967-f008:**
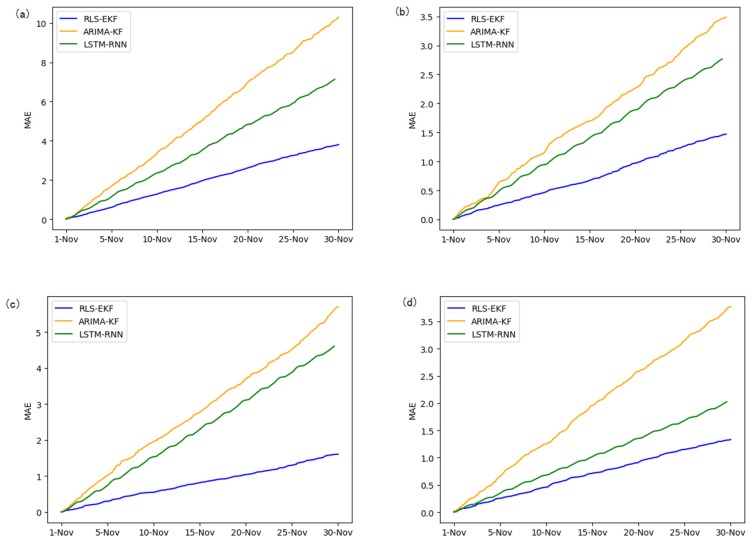
Mean absolute error (MAE) of 01_521 (**a**), 03_6479 (**b**), 04_6276 (**c**), and 06_28250 (**d**) at 15-min intervals.

**Figure 9 sensors-19-04967-f009:**
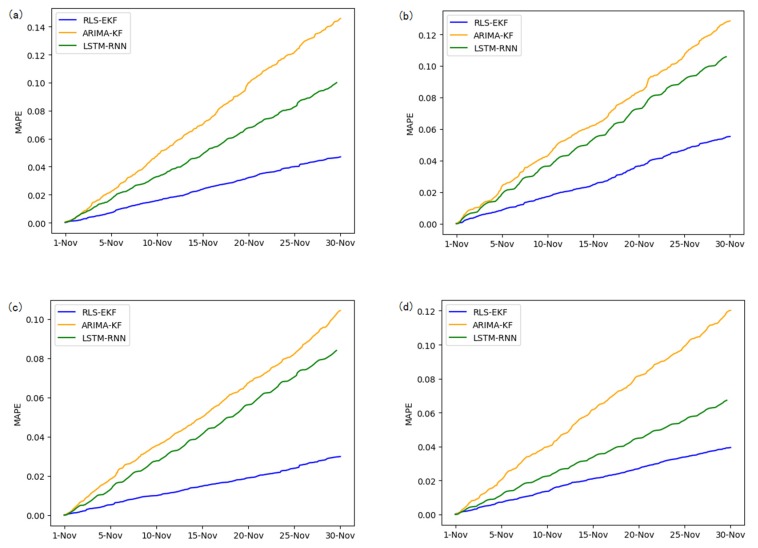
Mean absolute percentage error (MAPE) of 01_521 (**a**), 03_6479 (**b**), 04_6276 (**c**), and 06_28250 (**d**) at 15-min intervals.

**Figure 10 sensors-19-04967-f010:**
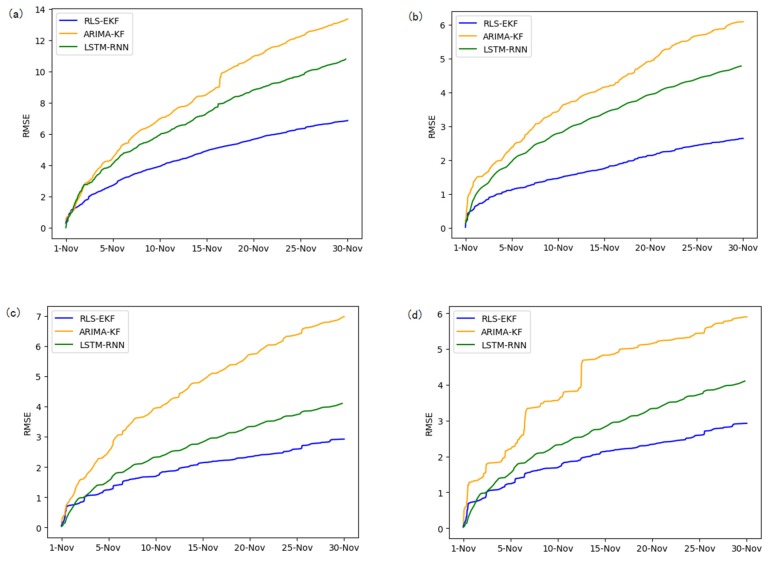
RMSE of 01_521 (**a**), 03_6479 (**b**), 04_6276 (**c**), and 06_28250 (**d**) at 30-min intervals.

**Figure 11 sensors-19-04967-f011:**
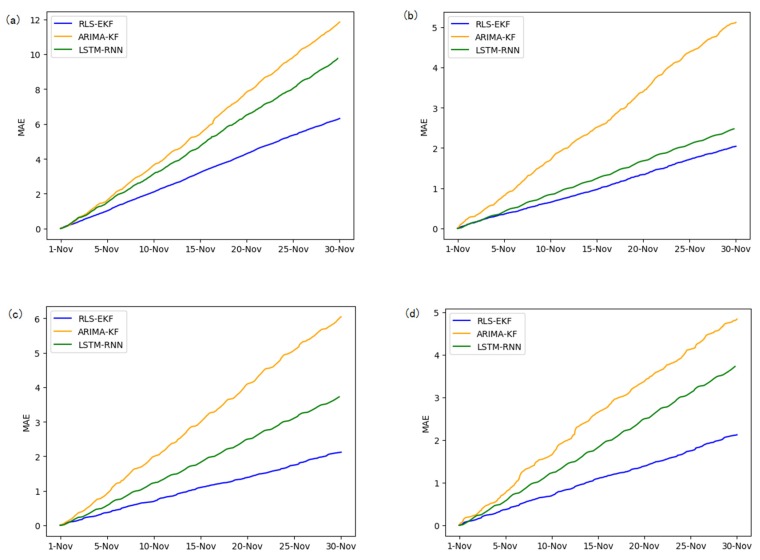
MAE of 01_521 (**a**), 03_6479 (**b**), 04_6276 (**c**), and 06_28250 (**d**) at 30-min intervals.

**Figure 12 sensors-19-04967-f012:**
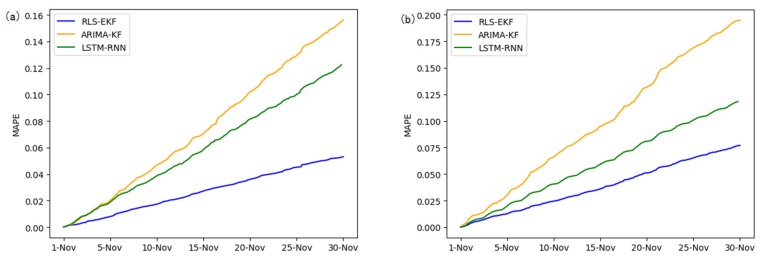
MAPE of 01_521 (**a**), 03_6479 (**b**), 04_6276 (**c**), and 06_28250 (**d**) at 30-min intervals.

**Figure 13 sensors-19-04967-f013:**
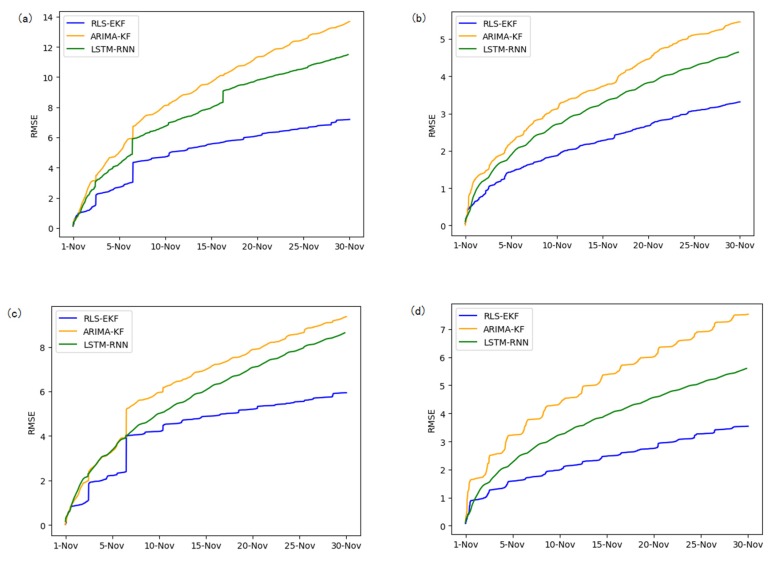
RMSE of 01_521 (**a**), 03_6479 (**b**), 04_6276 (**c**), and 06_28250 (**d**) at one-hour intervals.

**Figure 14 sensors-19-04967-f014:**
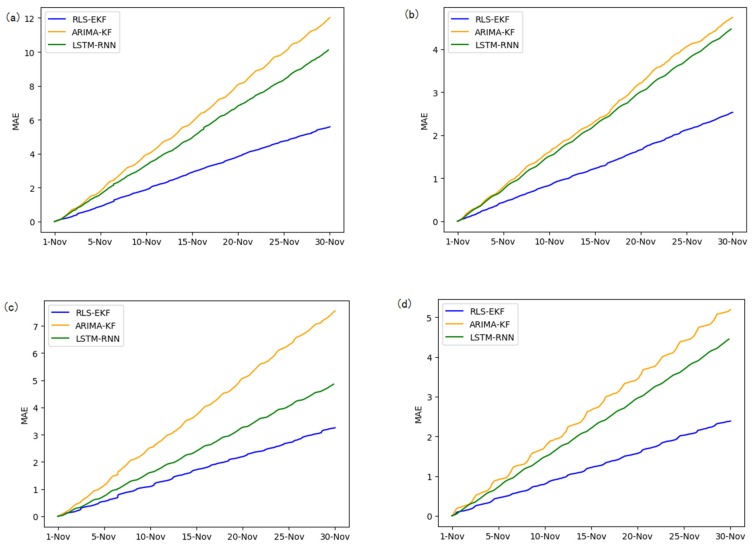
MAE of 01_521 (**a**), 03_6479 (**b**), 04_6276 (**c**), and 06_28250 (**d**) at one-hour intervals.

**Figure 15 sensors-19-04967-f015:**
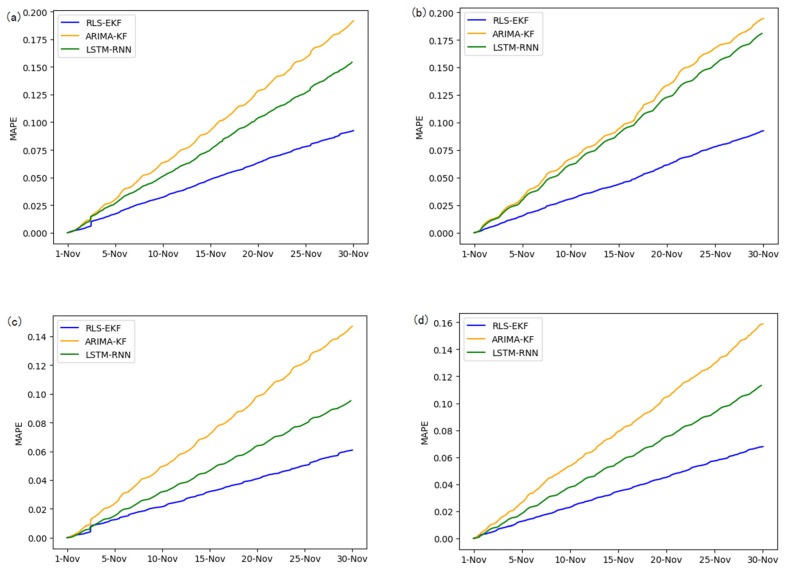
MAPE of 01_521 (**a**), 03_6479 (**b**), 04_6276 (**c**), and 06_28250 (**d**) at one-hour intervals.

**Table 1 sensors-19-04967-t001:** Pearson correlation coefficient of historical influence factors.

Timeslots	Seg. 1	Seg. 2	Seg. 3	Seg. 4
previous 1 h	0.4340	0.4398	0.4477	0.5475
previous 2 h	0.2865	0.3252	0.2934	0.3287
previous 3 h	0.2116	0.2545	0.2005	0.2289
previous 4 h	0.1683	0.2510	0.1472	0.1775
previous 5 h	0.2289	0.2223	0.1553	0.1599
previous 6 h	0.1050	0.0888	0.1309	0.1479
simultaneous timeslot of previous day	0.1969	0.1381	0.1368	0.1937
simultaneous timeslot of previous 2 days	–0.0483	–0.0557	–0.1026	–0.0888
simultaneous timeslot of previous 3 days	–0.1666	–0.1473	–0.1116	–0.1150
simultaneous timeslot of previous 4 days	–0.1711	–0.0473	–0.0499	–0.0307
simultaneous timeslot of previous 5 days	–0.0394	–0.0162	–0.0429	–0.0580
simultaneous timeslot of previous 6 days	0.1305	0.1193	0.0263	0.0979
simultaneous timeslot of previous 7 days	0.2015	0.1923	0.1353	0.1240

**Table 2 sensors-19-04967-t002:** The raw data format.

Item	Description
driver ID	desensitization
order ID	desensitization
timestamp	Unix epoch
latitude	dd.ddddd
longitude	ddd.ddddd
status	0: empty; 1: passenger; 2: parking

**Table 3 sensors-19-04967-t003:** Road segment information.

ID	Road Segment
01_521	Second Section of North Third Ring Road
03_6479	North Third Section of First Ring Road
04_6276	First Section of Hongxing Road
06_28250	Third Section of Jinxianqiao Road

**Table 4 sensors-19-04967-t004:** Quantized values of weather and date attributes.

Quantized Value	Weather Condition	Date Attribute
1	sunny, cloudy	first and last weekday
2	light rain, sleet	other weekdays
3	rain	weekend
4	heavy rain	first and last day of the holidays
5	snow, heavy snow	other holidays

**Table 5 sensors-19-04967-t005:** Performance comparison of different approaches for AS prediction.

Segment	Approach	15 min			30 min			1 h		
		RMSE	MAE	MAPE	RMSE	MAE	MAPE	RMSE	MAE	MAPE
01_521	ARIMA-KF	12.01	10.30	14.6%	13.36	11.84	15.6%	13.67	12.01	19.1%
	LSTM-RNN	7.81	6.87	9.1%	10.58	9.83	12.2%	11.15	10.43	15.8%
	RLS-EKF	4.35	3.81	4.7%	6.86	6.31	5.3%	7.19	5.58	9.2%
03_6479	ARIMA-KF	4.36	3.48	12.8%	6.09	5.12	19.5%	5.45	4.74	19.4%
	LSTM-RNN	3.83	2.82	10.9%	4.64	2.43	11.7%	4.67	4.52	18.2%
	RLS-EKF	1.95	1.46	5.5%	2.64	2.04	7.7%	3.31	2.53	9.2%
04_6276	ARIMA-KF	6.79	5.70	10.4%	6.98	6.05	11.7%	9.36	7.54	14.7%
	LSTM-RNN	5.84	4.52	8.5%	4.01	3.82	7.3%	8.33	4.69	9.1%
	RLS-EKF	2.24	1.61	3.0%	2.92	2.12	3.9%	5.94	3.25	6.1%
06_28250	ARIMA-KF	4.32	3.77	12.0%	5.90	4.84	13.7%	7.54	5.19	15.9%
	LSTM-RNN	2.26	2.02	6.8%	4.23	3.85	7.3%	5.21	4.63	11.2%
	RLS-EKF	1.76	1.33	3.9%	2.92	2.13	5.3%	3.54	2.39	6.8%

**Table 6 sensors-19-04967-t006:** RMSE of different influencing cases based on RLS-EKF.

Segment	Prediction Horizon	Miss-AS-hst	Miss-AS-ft	Miss-WC-ct	Miss-DA-ct	No Missing Factor
01_521	15 min	5.84	4.98	4.57	4.38	4.35
	30 min	7.69	7.21	7.02	6.89	6.86
	1 h	7.85	7.34	7.23	7.22	7.19
03_6479	15 min	3.05	2.88	2.60	2.52	1.95
	30 min	3.23	2.87	2.75	2.69	2.64
	1 h	3.92	3.48	3.41	3.36	3.31
04_6276	15 min	2.70	2.28	2.50	2.35	2.24
	30 min	3.51	3.03	3.01	2.95	2.92
	1 h	6.39	6.08	6.03	5.98	5.94
06_28250	15 min	2.41	1.89	1.83	1.80	1.76
	30 min	3.76	3.05	3.03	2.97	2.92
	1 h	3.92	3.64	3.62	3.61	3.54

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
