# Peer review of "FCG-ASpredictor: An Approach for the Prediction of Average Speed of Road Segments with Floating Car GPS Data"

_sensors, 2019, doi:10.3390/s19224967_

Round 1

Reviewer 1 Report

 This paper should be considered following comment during revision:

More detail on the adopted traffic data should be provided. The data is from DiDi chuxing Chengdu branch. As we know, the behavior of DiDi cars are different from the personal owned cars, or public buses. For example, a DiDi car will occasionally stop to pick up passengers during the trip. Therefore, it is not clear the AS prediction in this paper is particularly on DiDi cars or all the vehicles in the transportations. In addition, it is interesting to discuss the adopted criteria on selecting the road segments. As discussed, there are four road segments selected. Are those segments the ones with less traffic lights? or the main road? What happen if we choose the road segment having lots of pedestrians and crossroads? The datasets of the 30 days that ranged from November 1–30, 2016 are assessed in this paper. However, this paper better considers the datasets with more samples collected in more days, say 180 days. This is because, the weekends, raining days etc. within 30 days will be rare and the prediction will be overfitting with respect to such limited datasets. For example, season as one important factor is ignored in this dataset. Also it uses the data from 01/11/2016-26/11/2016 as the training sample, and 27/11/2016-30/11/2016 as the forecast sample, it is highly likely that the prediction accuracy will be jeopardized by random noises. It is difficult to figure out the benefit of RLT-EKF over the benchmark methods in Fig.7-9.  The y-axis is suggested to be used to denote the RSME, MAE, MAPE of each method, instead of the average speed directly. Also please provide markers in each line to differentiate the considered methods. Please proof read the paper to avoid the typos, like the on “Aata” in line 112. Also please improve the quality of the Fig.5 and Fig.6.

Author Response

Dear Reviewer,

Thanks very much for taking your time to review this manuscript. I really appreciate all your comments and suggestions!

Please see the attachment of my responses.

Reviewer 2 Report

This paper proposed an approach for predicting average speed of road segment using floating car data. Overall,an approach integrated a spatiotemporal correlation calculation method  with a recursive least squares–extended Kalman filter is proposed, and evaluated using floating car data. However, the basic idea in the proposed approach is not new, this paper is not ready, and must be improved.

the average speed of road segment is calculated as the average value of instantaneous speeds in sample. I disagree with this calculation. two reasons:1). the average speed of road segment usually refers to the average speed of travel through the road segment, travel speed is very different with instantaneous speed;2). the instantaneous speeds are not evenly on an urban road due to signal control, and the distribution of instantaneous speed on an urban road depends on the length, geometric layout and signal timings on that road. what is the penestration rate and sampling frequency of floating car data? How can the authors trust the data without those information? where does the groud truth average speed come from? the english writing is very poor, a proofreading is recommended. And several figures are blurry and don't meet format requirements.

Author Response

(The authors gave the same response as above.)

Reviewer 3 Report

The paper provides a method for prediction of segment average speed using Karman filtering.

I consider that in the age of Deep Learning the use of Recursive Least Squares and a Kaman filter is not a very original purpose.

Nevertheless, the practical nature of the project, using real data captured from actual streets in China, is valuable. And the results are interesting compared with LSTM networks and ARIMA models.

Paper is well presented and with moderate interest.

Author Response

Dear Reviewer,

Thanks very much for taking your time to review this manuscript. I really appreciate all your comments and suggestions!

We will work hard on the latest technology research, such as Deep learning.

Round 2

Reviewer 1 Report

The authors have addressed all my comments.

Reviewer 2 Report

I agree the authors' revision.